# Omicron sub-lineage BA.5 infection results in attenuated pathology in hACE2 transgenic mice

Zaigham Abbas Rizvi [1,2✉], Jyotsna Dandotiya[1], Srikanth Sadhu[1,2], Ritika Khatri[3], Janmejay Singh[4], Virendra Singh[1,2], Neeta Adhikari[1], Kritika Sharma[1,2], Vinayake Das[1,2], Amit Kumar Pandey [5], Bhabatosh Das [6], Guruprasad Medigeshi [4], Shalendra Mani [3], Shinjini Bhatnagar [7], Sweety Samal[3], Anil Kumar Pandey[8], Pramod Kumar Garg[9] & Amit Awasthi [1,2✉]

A recently emerged sub-lineage of Omicron, BA.5, together with BA.4, caused a fifth wave of coronavirus disease (COVID-19) in South Africa and subsequently emerged as a pre-dominant strain globally due to its high transmissibility. The lethality of BA.5 infection has not been studied in an acute hACE2 transgenic (hACE2.Tg) mouse model. Here, we investigated tissue-tropism and immuno-pathology induced by BA.5 infection in hACE2.Tg mice. Our data show that intranasal infection of BA.5 in hACE2.Tg mice resulted in attenuated pulmonary infection and pathology with diminished COVID-19-induced clinical and pathological mani-festations. BA.5, similar to Omicron (B.1.1.529), infection led to attenuated production of inflammatory cytokines, anti-viral response and effector T cell response as compared to the ancestral strain of SARS-CoV-2, Wuhan-Hu-1. We show that mice recovered from B.1.1.529 infection showed robust protection against BA.5 infection associated with reduced lung viral load and pathology. Together, our data provide insights as to why BA.5 infection escapes previous SARS-CoV-2 exposure induced-T cell immunity but may result in milder immuno-pathology and alleviated chances of re-infectivity in Omicron-recovered individuals.

[1] Centre for Immuno-biology and Immunotherapy, Translational Health Science and Technology Institute, NCR-Biotech Science Cluster, 3rd Milestone, Faridabad-Gurgaon Expressway, Faridabad, Haryana 121001, India. [2] Immunology-Core Lab, Translational Health Science and Technology Institute, NCR-Biotech Science Cluster, 3rd Milestone, Faridabad-Gurgaon Expressway, Faridabad, Haryana 121001, India. [3] Centre for Viral Therapeutics and Vaccines, Translational Health Science and Technology Institute, NCR-Biotech Science Cluster, 3rd Milestone, Faridabad-Gurgaon Expressway, Faridabad, Haryana 121001, India. [4] Bioassay Laboratory, Translational Health Science and Technology Institute, NCR-Biotech Science Cluster, 3rd Milestone, Faridabad-Gurgaon Expressway, Faridabad 121001, India. [5] Centre for Tuberculosis and Bacterial Diseases Research, Translational Health Science and Technology Institute, NCR-Biotech Science Cluster, 3rd Milestone, Faridabad-Gurgaon Expressway, Faridabad, Haryana 121001, India. [6] Centre for Microbiome and Anti-Microbial Resistance, Translational Health Science and Technology Institute, NCR-Biotech Science Cluster, 3rd Milestone, Faridabad-Gurgaon Expressway, Faridabad, Haryana 121001, India. [7] Centre for Maternal and Child Health, Translational Health Science and Technology NCR-Biotech Science Cluster, 3rd Milestone, Faridabad-Gurgaon Expressway, Faridabad, Haryana 121001, India. [8] Department of Physiology, ESIC Medical College & Hospital, Faridabad 121001, India. [9] Department of Gastroenterology, All India Institute of Medical Sciences, New Delhi 110029, India. ✉email: zaigham.abbas15@gmail.com; aawasthi@thsti.res.in

The Omicron (Pango lineage B.1.1.529) variant of severe acute respiratory syndrome coronavirus 2 (SARS-CoV- 2) was first reported in November 2021 almost 2 years after the first case of COVID-19 was reported in Wuhan, China. While the ancestral strain of SARS-CoV-2, Wuhan-Hu-1, caused the first wave, the second wave of COVID-19 was caused by the Delta (B.1.617.2) strain of SARS-CoV-2[1–4]. Soon after the identification of Omicron (B.1.1.529) strain of SACS-CoV-2 in South Africa, World Health Organization (WHO) included B.1.1.529 variant as a Variant of concern (VoC), which subsequently quickly replaced Delta (B.1.617.2) strain and become the predominant circulating strain of SARS-CoV-2 across the globe. The spread of Omicron posed a greater challenge due to its ability to evade vaccine-derived immunity especially antibody-mediated neutralization[5]. Mutational characterization of Omicron revealed that there were more than 30 mutations in its spike protein, which were found to be resulted in compromised immunity in COVID-19-vaccinated individuals[1,4,6,7]. Ever since its identification, Omicron variant of SARS-CoV-2 continues to evolve and leading to the generation of at least five major sub-lineages that are characterized by different sets of mutations[8]. The first three sub-lineages of Omicron were named as BA.1 BA.2 and BA.3, which resulted in the fourth epidemic wave in South Africa due to their high degree of transmissibility. Epidemiological data suggested that BA.1 BA.2 and BA.3 sub-lineages of Omicron caused milder infection with a reduced risk of hospitalization[9,10]. After the first three sub-lineages, BA.4 and BA.5 sub-lineages of Omicron were emerged and identified as new sub-lineages. Both BA.4 and BA.5 harbored substitution of L with S amino acid at 452 positions of spike (S) protein[8], which makes BA.4 and BA.5 more transmissible than its lineage Pango ancestor[9]. However, the disease severity caused by BA.5, as compared to ancestral, strain of SARS-CoV-2 especially in terms of multi-organ infectivity associated with severe COVID-19 pathology and mortality has not been studied.

Phylogenetically, BA.2 sub-lineage is the closest to both BA.4 and BA.5 sub-lineage of Omicron, suggesting that BA.4 and BA.5 may have originated from BA.2 by acquiring additional mutations[8]. Although BA.4 and BA.5 share identical spike protein, both BA.4 and BA differ from BA.2 with the following mutations: 69–70 deletion, and two substitutions L542R, F486V and have a reverse mutation of wild-type amino acid at Q493 position[8,9]. In addition, BA.5 has also accumulated mutations outside spike protein and contains M:D3N mutation which is absent in both BA.4 and BA.2. It also has additional reversions at ORF6:D61 and nucleotide positions 26,858 and 27,259[8,9]. All these accumulated mutations in BA.5 may have contributed to its immune evasion mechanism. which could potentially leading to BA. 5-driven surge of infection across the Europe and United States of America. BA.5-driven wave of COVID-19 had replaced BA.4 strain of SRAS-CoV-2 and made BA.5 the most predominant circulating strain of SARS-CoV-2[11]. Though BA.4 and BA.5 share a similarity in their spike protein, it is unclear as to why the transmission rate of BA.5 is higher than BA.4 strain of SARS-CoV-2. It was previously identified that Omicron sub-lineage of SARS-CoV-2 were found to be associated with an attenuated antibody-mediated protection from vaccination-derived immunity. However, there is insufficient data on BA.5-driven disease severity in previously SARS-CoV-2 infected but recovered individuals. Moreover, the T cell response, which plays a crucial role in protective immunity against COVID-19, remains unexplored in BA.5 infection. In this study, we used hACE2.Tg mice model to study the immune-pathological changes induced by BA.5 infection and compared them with the ancestral and its parental lineage B.1.1.529 infection. We identified that hACE2.Tg mice infected with BA.5 showed diminished clinical signs with

decreased viral load in the lungs as compared to the ancestral SARS-CoV-2 infection. Notably, as compared to the ancestral strain, BA.5 infection disseminated to the other vital organs and resulted in elevated viral load in the brain, spleen and kidney. Immunopathological data indicated that BA.5 infection, as compared to the ancestral strain, leads to an overall attenuated inflammatory response. Moreover, restimulation of splenocytes with wild-type RBD and spike-proteins resulted in the diminished frequency of IFNγ + T helper (Th) cells. Notably, animals recovered from B.1.1.529 infection were protected against BA.5 re-infection with reduced pulmonary viral load and pathology. Together, our findings highlight the following three major insights into BA.5 infection pathology by using hACE2.Tg mice: 1) BA.5 infection induces attenuated immunopathology, 2) BA.5 infection leads to an elevated tissue tropism to extra-pulmonary organs, and 3) attenuated T cell response with improved protection in B.1.1.529 recovered hACE2.Tg mice.

## Results

**BA.5 infection in hACE2.Tg mice result in attenuated pulmonary viral load and pathology.** Both hamster and hACE2.Tg mice were conventionally used for studying the pathological changes induced by SARS-CoV-2 infection. Hamsters have been described as a natural model which mimics mild to moderate COVID-19 pathology without causing severe respiratory distress, as seen in a small but significant number of clinical cases[12–16]. hACE2.Tg mice were previously developed for studying SARS-CoV infection and pathology, and subsequently adapted for studying the acute infection of SARS-CoV-2[16,17]. Although the BA.5 infection and associated pathology were reported in hamsters, there is still a lack of data on BA.5 infection in hACE2.Tg mice model which mimics severe COVID-19 pathology[18]. Here, we used hACE2.Tg mice to investigate pulmonary and extra-pulmonary manifestations following BA.5 infection and compared it with ancestral and parental strain, B.1.1.529, of SARS-CoV-2. BA.5 infection in hamsters leads to a marginal but non-significant reduction in body mass[18]. In contrast, hACE2.Tg mice infected with BA.5 resulted in a significant reduction of body mass as compared to uninfected (UI) hACE2.Tg mice on 6 days post-infection (dpi). However, the overall percentage decrease in the change of body mass of BA.5-infected mice as compared to its day 0 body mass remains negligible (Fig. 1a). The overall body weight changes of BA.5-infected animals were found to be similar to B.1.1.529 infected mice. BA.5 infection resulted in a significantly attenuated pathology with zero mortality in hACE2.Tg mice up to 14 dpi as compared to ancestral Wuhan-Hu-1 infection which resulted in 100% mortality by 7 dpi (Fig. 1b).

Next, we used lung samples at 6 dpi of hACE2.Tg mice infected with BA.5, B.1.1.529 or ancestral strain for comparative gross pathological examination (Fig. 1c). As indicated by HE staining, lungs of both B.1.1.529 and BA.5-infected showed significantly attenuated pathological scores as compared to the ancestral strain (Fig. 1c and Supplementary Fig. 1a). Further histopathological evaluation associated with BA.5 infection show significantly reduced signs of pneumonitis, inflammation, lung injury, alveolar epithelial injury and overall disease index score as compared to ancestral strain infection (Fig. 1d and Supplementary Fig. 1b). To understand if the attenuated pathology in BA.5-infected animals was due to reduced viral entry and replication in the lungs, we carried out immunohistochemistry (IHC) for N-antigen of SARS-CoV-2 and viral load estimation from lung and trachea samples. Our data suggest that as compared to ancestral strain, there is a 20–25% reduction in N-antigen IHC score in the lungs of BA.5 and parental lineage B.1.1.529 variant (Fig. 1e, f). Furthermore, TCID50 data indicates that both ancestral Wuhan and BA.5

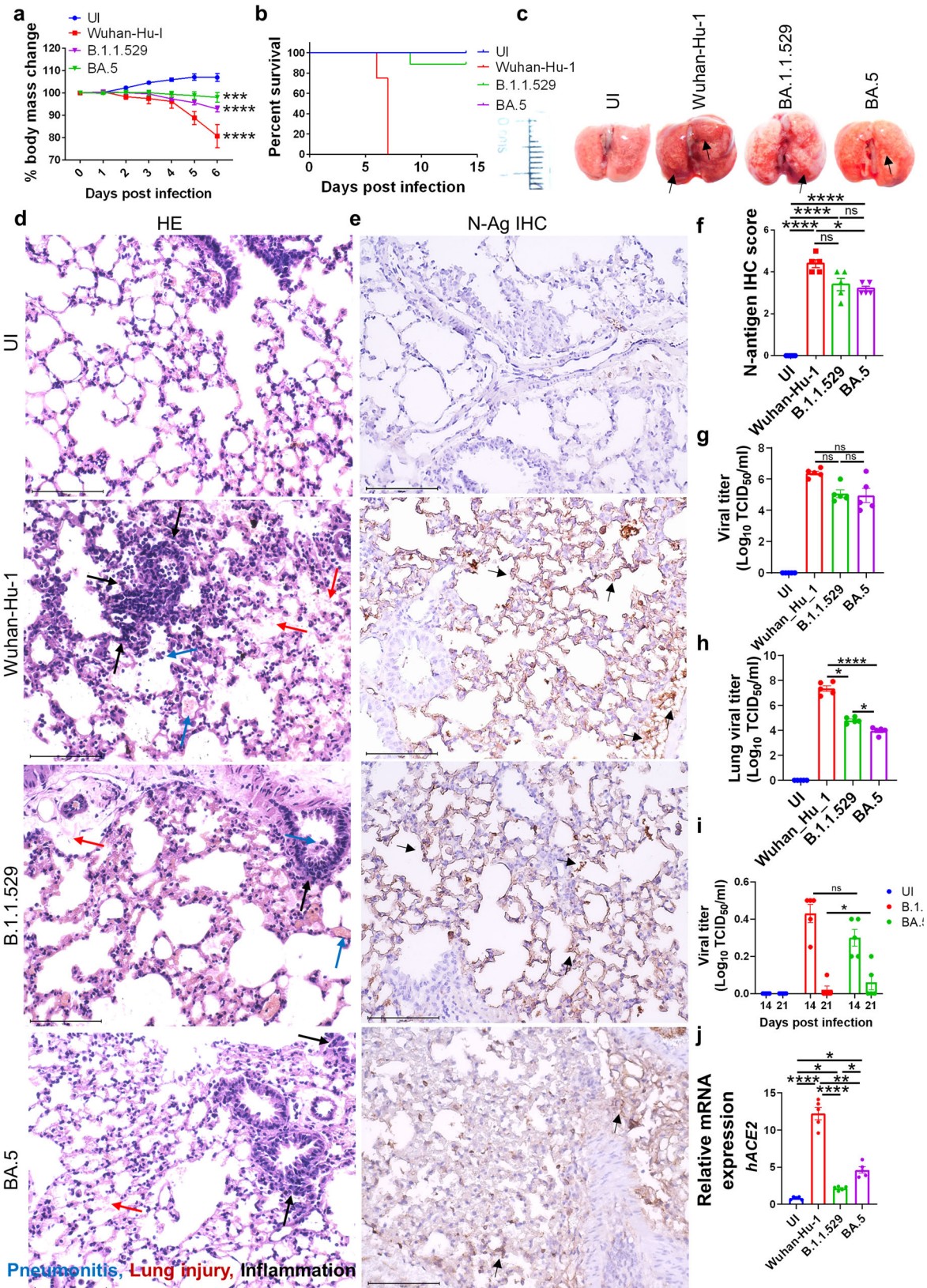

strain showed similar viral load in trachea, however, the lung viral load of BA.5 infected mice was significantly lower with 3-4 $\log_{10}$ reduction as compared to ancestral Wuhan strain suggesting that BA.5 preferably infects the upper respiratory tract (Fig. 1g, h). We further monitored the lung viral load longitudinally at 14 and 21 dpi. Our data shows that most of the BA.5 infection is cleared

from the lungs by day 14 with little or no viral titer at 21 dpi (Fig. 1i).

The entry of the SARS-CoV-2 into host cells is primarily driven by three entry factors: angiotensin-converting enzyme-2 (ACE2), transmembrane serine protease (TMPRSS2), neuropilin-1 (NRP-1)[19–21]. We found that BA.5 infection resulted in significant

**Fig. 1 BA.5 infection in hACE2.Tg mice results in attenuated pulmonary viral load and pathology.** Mice were challenged with ancestral Wuhan strain (Wuhan-Hu-1), Omicron (B.1.1.529) or omicron-sub-lineage (BA.5) strain intranasally with $10^5$ pfu virus/ mice. Clinical signs and body weight were recorded for 6 days post challenge or until they become moribund for immunopathological studies. For survival study animals were monitored till day 14 post infection. **a** line graph showing mean percentage changes in body mass of the mice as compared to the day 0 body mass. **b** Percentage survival of mice till day 14 after different challenge strains. **c** Representative images of harvest lungs on day 6 post infection showing regions of inflammation & pneumonitis (black arrow). **d** Representative H & E stained images of lung section at 40X showing pneumonitis (blue arrow), lung injury (red arrow), inflammation (black arrow). **e** Representative IHC images for N antigen (black arrow) in the lung sections (**f**) blinded scoring of IHC stain by trained pathologist on the scale of 0 to 5 (where 0 meant no stain & 5 meant highest brown color stain distribution). **g** Viral titer in the trachea expressed as TCID50. **h** Viral titer in the lungs expressed as TCID50. **i** longitudinal viral titer in the lungs expressed as TCID50 at day 14 and 21 post infection (**j**) relative mRNA expression of hACE2 gene in the mice lungs. For all experiment $n = 5$. One way-Anova using non-parametric Kruskal–Wallis test for multiple comparison. ns non-significant, $*P < 0.05$, $**P < 0.01$, $***P < 0.001$, $****P < 0.0001$.

upregulation of hACE2 mRNA expression in the lungs as compared to both UI and its parental lineage B.1.1.529 infection (Fig. 1j). However, the mRNA expression of hACE2 in BA.5 infection remained significantly low as compared to ancestral strain infection (Fig. 1j). Notably, the mRNA expression of TMPRSS2 and NRP-1 in BA.5 infected lungs were similar to UI with no significant changes (Supplementary Fig. 1c). Together, assessment of pulmonary pathology induced by BA.5 infection in hACE2.Tg mice revealed overall attenuated pathology with a decreased viral load which was comparable to parental lineage B.1.1.529 infection.

**BA.5 infection induced milder inflammatory response in hACE2.Tg mice.** SARS-CoV-2 infection leads to pulmonary manifestations which include, but not limited to, cytokine release syndrome (CRS), cellular injury, and respiratory distress[22–24]. Our data on BA.5-mediated pulmonary pathology in hACE2.Tg mice corroborated with previous findings in which BA.5 infection in the hamster caused a milder pulmonary pathology. However a detailed characterization of immunopathological changes post SAR-CoV-2 infection in hamsters is limited due to lack of available reagents. Therefore, we used hACE2.Tg mice to better understand and characterize the immunopathological changes associated with BA.5 sub-lineage infection. CRS is a systemic release of inflammatory cytokines following SARS-CoV-2 infection, which includes the release of proinflammatory cytokines like interferon-gamma (IFN-γ), tumor necrosis factor-alpha (TNF-α), interleukin (IL)−4, IL-6, IL-17A, IL-33, etc[22,25,26]. We used qRT-PCR to access the mRNA expression of these cytokines in the lung samples of BA.5-infected hACE2.Tg mice and compared them with ancestral and B.1.1.529 infected hACE2.Tg mice. Our data shows that, as compared to ancestral Wuhan-Hu-1 strain, BA.5 infection in hACE2.Tg mice shows an overall decrease in the mRNA expression of IL-4, IL-6, IL-17A, and IL-33 (Fig. 2a). However, there was approximately two folds increase in mRNA expression of IFNγ and TNF-α in BA.5 infection when compared to its parental B.1.1.529 infection (Fig. 2a). Next, we carried out intracellular cytokine staining from the cells of the bronchoalveolar lavage fluid (BALF) in BA.5-infected hACE2.Tg mice. Our data show that cells from BALF of BA.5 infected hACE2.Tg mice show an overall lower frequency of IFN-γ, IL-17A, IL-10-producing CD4+ T cells as well as IFN-γ, IL-10-producing CD8+ T cells, as compared to the ancestral Wuhan-Hu-1 infection. However, BA.5 and its parental strain, B.1.1.529, infection leads to the similar frequency of IFN-γ, IL-17A, IL-10-producing CD4+ T cells as well as IFN-γ, IL-10-producing CD8+ T cells in BALF (Fig. 2b & Supplementary Fig. 1d, e). In addition, the frequency of eosinophils and mast cells in BALF was found to be lower in BA.5, as compared to the ancestral Wuhan-Hu-1 infection in hACE2.Tg mice (Fig. 2b & Supplementary Fig. 1d, e). Interestingly, the frequency of eosinophils and mast cells was found to be elevated in BA.5 infected as compared to its parental strain B.1.1529 infection (Fig. 2c).

Next, to study the host anti-viral response in BA.5 infection, we carried out mRNA expression of anti-viral genes and interferon stimulatory genes (ISGs), which get activated post-sensing of viral RNA by toll-like receptors (TLRs) or rig-like receptor (RLRs). We found a significant increase in the lung mRNA expression of IFN-α, and IFN-β in BA.5 infection which was found to be similar to the ancestral Wuhan-Hu-1 infection. Notably, interferon regulatory factor-1 (IRF-1), which regulates the expression of interferon genes, and IFN-induced transmembrane (IFITM) mRNA expression showed a significant increase in BA.5, as compared to ancestral Wuhan-Hu-1, infection (Supplementary Fig. 1f). While the relative mRNA expression of adenosine deaminase RNA-1 specific (ADAR-1) and latent RNase (RNaseL) did not show any difference between BA.5 and ancestral Wuhan-Hu-1 infection of lungs samples of hACE2.Tg mice (Supplementary Fig. 1f), mRNA expression of anti-viral genes 2'−5'-oligoadenylate synthetase (OAS)−2 and OAS-3, OAS-19 was found to be significantly up-regulated in BA.5 as compared to ancestral Wuhan-Hu-1 infection. The mRNA expression of OAS-2, OAS-19 and mx-1 (interferon-induced GTP-binding protein) was found to be similar between BA.5 and B.1.1.529 infected lungs of hACE2.Tg mice. However, there was an approx. a 2-fold decrease in the expression of mx-2 in BA.5 infection lungs as compared to B.1.1.529 infection (Supplementary Fig. 1g).

We next accessed the immunological changes induced by BA.5 infection in the spleen that reflects the systemic immune response upon infection. Our immunophenotyping data from splenocytes show that the frequency of CD4+ T cells and CD8+ T cells in BA.5 infected mice were similar to that of parental B.1.1529 infection[27–29] (Supplementary Fig. 2a). Moreover, the frequency of other immune cells populations such as γδ+ T cells, NK cells, monocytes, neutrophils, MDSCs, and macrophages remained similar between BA.5-infected and B.1.1529-infected hACE2.Tg mice. However, a 6-fold decrease in the frequency of NKT cells was observed in BA.5, as compared to parental B.1.1.529, infected hACE2.Tg mice (Supplementary Fig. 2b–e). In addition, we evaluated the mRNA expression of cytokines genes in the splenocytes by qPCR. We observed relatively lower mRNA expression of inflammatory cytokines (IFN-γ, TNF-α, IL-4, IL-6, IL-17A and IL-33) in BA.5 infection as compared to ancestral Wuhan-Hu-1 infection (Supplementary Fig. 3a). In line with this, the percentage frequency of CD4+ T cells that produce IFN-γ, IL-17 or IL-10 in the splenocytes remained low in BA.5 infection as compared to ancestral Wuhan-Hu-1, but were found to be comparable to that of parental B.1.1.529 infected mice (Supplementary Fig. 3b, c). Moreover, the mRNA expression of anti-viral and ISG genes was also similar to the parental B.1.1.529 infection but significantly attenuated as compared to ancestral Wuhan-Hu-1 infected mice (Supplementary Fig. 3d, e). We had previously shown that ancestral SARS-CoV-2 infection results in splenic atrophy in hACE-2.Tg mice[13]. However, while significant splenic atrophy was observed post ancestral strain, we did not find any

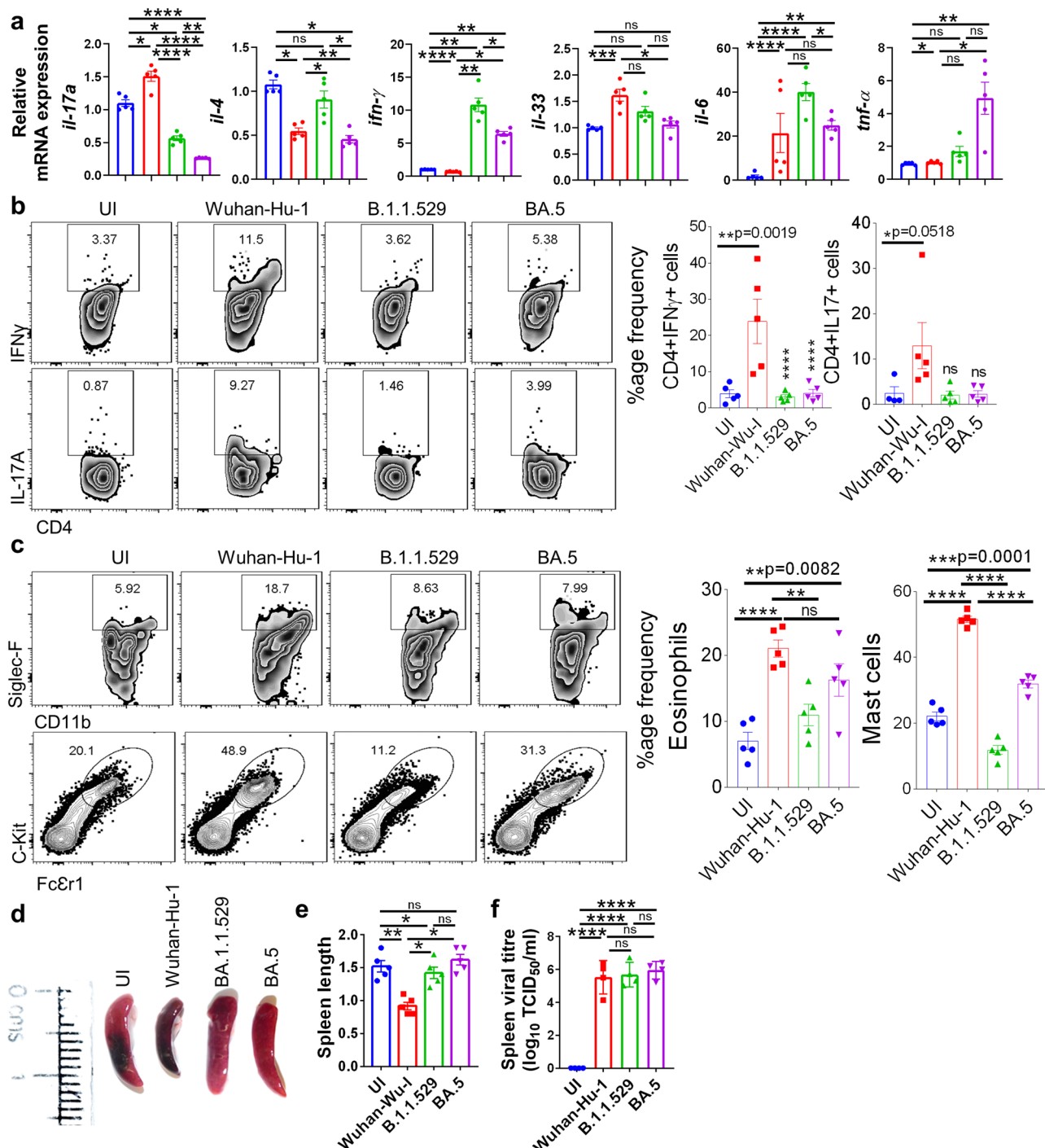

**Fig. 2 BA.5 infection induced milder inflammatory response in hACE2.Tg mice. a** mRNA expression of inflammatory cytokines genes were studied by qPCR from day 6 lung samples and plotted as bar graph mean ± SEM. **b** Dot plots and bar graphs showing %age frequency of IFN-γ or IL-17A producing CD4+ T cells in the bronchio-alveolar lavage fluid (BALF) or infected (I)/ uninfected mice (UI). **c** Dot plots and bar graphs showing % age frequency of Eosinophils (SiglecF+CD11b+) & mast cells (ckit+FcƐr1+) cells in the BALF (**d** and **e**) length of excised spleen (**f**) spleen viral load. For all experiment $n = 5$. One way-Anova using non-parametric Kruskal–Wallis test for multiple comparison. ns non-significant, *$P < 0.05$, **$P < 0.01$, ****$P < 0.0001$.

signs of splenic atrophy post BA.5 and parental B.1.1.529 infection in hACE.Tg mice (Fig. 2d, e). Notably, there was a roughly 2 fold increase in the spleen viral load in BA.5 group as compared to ancestral Wuhan-Hu-1 infected animals which prompted us to investigate the expression of virus entry factors in the spleen (Fig. 2f). The mRNA expression of viral entry factors in the spleen remained relatively low and comparable to the mRNA expression of UI samples (Supplementary Fig. 3f).

We also evaluated the immunological changes in the draining lymph nodes (dLN) of the lungs. Our data shows that the percentage frequency of the major immune cell population in the dLN of BA.5 infected mice remained similar to that of B.1.1.529 infected mice. Moreover, the frequency of IFNγ+, IL10+ and IL-17A-producing T cells in the dLN of BA.5 infected mice were similar to parental B.1.1.529 infected mice (Supplementary Fig. 4f–h). Taken together, the inflammatory profile and mRNA

expression pattern in BA.5 infection remained more or less similar to that of B.1.1.529 infection but was significantly attenuated when compared to ancestral Wuhan-Hu-1 infection.

**Antigen-specific humoral and T helper cell response in BA.5 infected mice**. With the rise of Omicron (B.1.1.529) variant infection, one of the major concerns was an ineffective neutralizing antibody response against B.1.1.529 infection in vaccinated or previously exposed individuals[2,10]. Emerging literature suggests that a compromised neutralizing antibody response against B.1.1.529 infection is due to acquired mutations in the spike protein[6,10,30–32]. Moreover, BA.5 shows further attenuation in neutralizing antibody response as compared to its parental lineage B.1.1529, which could be attributed due to the additional mutations in the spike protein of BA.5[6,8–10,32]. However, whether the anti-viral Th1 response in individuals vaccinated with spike protein-based vaccines is sufficiently preserved to provide protection against BA.5 is largely unknown. To evaluate the antigen-specific Th1 cell response following BA.5 infection, we antigenically (wild type RBD protein or wild type spike protein) re-stimulated splenocytes from BA.5 infected mice (6 dpi) in-vitro. Although there were no difference was observed in the frequency of IL-2-producing $CD4^+$ or $CD8^+$ T cells in BA.5, as compared to B.1.1.529, infected mice in response to restimulation with wild type RBD and spike protein (Fig. 3a, b, e, f). However, we observed a, 3-4 folds reduction in the frequency of $IFN\gamma^+CD4^+$ and about 1.5-2 folds reduction in the frequency of $IFN\gamma^+CD8^+$ T cells in response to restimulation with wild-type RBD and spike protein from BA.5, as compared to ancestral Wuhan strain, infected mice (Fig. 3c, d, g, h). In addition, we also evaluated SARS-CoV-2-specific humoral response by determining the anti-RBD and anti-spike IgG antibodies in the serum samples of infected mice at indicated time points post-infection. Although both anti-RBD and anti-spike IgG antibodies showed a good titer at day 14 post-infection, only anti-spike IgG antibody titer was found to be maintained even at day 21 post-infection with a marginal decrease in anti-RBD IgG antibody titer (Fig. 3i, j). Together, we show an overall diminished T and B cell immunity against BA.5 as compared to ancestral strain infection.

**BA.5 infection of extra-pulmonary organs**. Ancestral Wuhan-Hu-1 infection was characterized by virus dissemination to other extra-pulmonary organs leading to infiltration of inflammatory cells and cause inflammation-induced dysfunction of the organs and/or multiple organs failure[12,13,33–36]. To understand the pathological changes manifested by BA.5 sub-lineage infection in major extrapulmonary organs of hACE2.Tg mice, we carried out viral load estimation with a detailed histopathological analysis of the vital organs. We found a surprisingly high viral load, as indicated by TCID50 in brain, kidney and colon of the mice infected with BA.5 variant which was significantly higher than the ancestral Wuhan-Hu-1 viral load (Fig. 4a, d, e respectively). However, a significantly reduced viral load was found in the heart of BA.5, as compared to Wuhan-Hu-1, infected hACE2.Tg mice. We did not find any difference in the viral load between BA.5 and Wuhan-Hu-1 infected livers (Fig. 4b, c respectively). In line with this, HE-stained sections of brain and kidney of BA.5 infected mice showed higher infiltration of immune cells and elevated pathological scores as compared to parental B.1.1.529 infection (Fig. 4f, i). The pathological assessment of heart, liver and colon showed marginal or no pathological score in BA.5 infected mice as compared to parental B.1.1.529 infection (Fig. 4g, h, j & Supplementary Fig. 5a–e). Together, our data points to differential viral dissemination and surprisingly BA.5 accumulation was found to be higher in brain, kidney and colon.

**Prior infection with Omicron protects hACE2.Tg mice from BA.5 challenge**. Pango lineage B.1.1.529 led to a global surge in infection with mild COVID-19 symptoms[7,37]. We wanted to test whether prior infection with Omicron provide protection against BA.5 infection in hAce2.Tg mice. We found that mice sera of 14 days post-infection (which could be achieved only in sublethal infection by Omicron in hACE2.Tg mice but not with Wuhan-Hu-1 infection), but not day 7, showed a good titers of anti-RBD and anti-spike antibody (Fig. 3i). Therefore, we reasoned that 14-day Omicron recovered hACE2.Tg mice would be better for understanding the disease pathology of re-infection with SARS-CoV-2 variants. We used this model for subsequent reinfection with different variants of SARS-CoV-2: Wuhan-Hu-1, B.1.617.2, B.1.1.529 and BA.5 variants (Fig. 5a). hACE2.Tg mice challenged with Wuhan-Hu-1 or B.1.617.2 strain succumb to death between 6–8 dpi (Fig. 5 b). However, robust protection was observed in Omicron-primed mice followed by challenges with Delta variant B.1.617.2 (Omi-B.1.617.2) as compared to ancestral Wuhan-Hu-1 (Omi-Wuhan-Hu-1) rechallenge (Fig. 5a, b). Omicron-infected mice showed marginal or no weight loss, no signs of mortality, and a lesser pathological score in the lungs when challenged with B.1.617.2 variant (Fig. 5b–e). In line with this, the histopathological assessment of HE-stained lung sections of rechallenged mice showed robust protection in mice rechallenged with Delta variant B.1.617.2 (Omi-B.1.617.2) with significantly attenuated histopathology as compared to ancestral Wuhan-Hu-1 (Omi-Wuhan-Hu-1) rechallenged mice (Supplementary Fig. 6a). Moreover, there was a reduced lung viral load in Omi-B.1.617.2 mice as compared to the Omi-Wuhan-Hu-1 ancestral strain rechallenge animals as confirmed by both qPCR and IHC (Fig. 5f & Supplementary Fig. 6b). Consistently, the mRNA expression of IL-6 was also found to be significantly reduced in animals reinfected with B.1.617.2 as compared to Wuhan-Hu-1 reinfected animals. However, mRNA expression of IFNα, IFNβ and IFNγ showed a non-significant difference in animals reinfected with B.1.617.2 as compared to Wuhan-Hu-1 reinfected animals. (Fig. 5g & Supplementary Fig. 6e). On the other hand, as expected rechallenge with B.1.1.529 or BA.5 produced no clinical manifestations of COVID19 in terms of weight loss, percentage survival and pathological score of excised lungs (Fig. 5h–k). In line with this, there was significantly reduced lung pathology observed in both Omi-B.1.1.529 and Omi-BA.5 rechallenge group (Supplementary Fig. 6c). Notably, the rechallenged group showed significantly reduced viral load in the lungs and low levels of inflammatory cytokine IL-6 and IFNγ mRNA expression levels. However, little or no changes in IFNα and IFNβ mRNA expression levels were observed between BA.5 single infection and BA.5 reinfection gorup (Fig. 5l, m & Supplementary Fig. 6d–f). Together, through our rechallenge study we provide direct animal model-based evidence showing low viral load and clinical pathologies as the basis of milder severity due to re-infection by emerging variants (Fig. 6).

## Discussion

The recent surge in SARS-CoV-2 infection, in the month of Jan-Feb, 2022, was mostly driven by BA.4 or BA.5 variants. Both BA.4 or BA.5 variants were identified as sub-lineage variants of B.1.1.529 Omicron strain[8]. Emerging evidences suggest that the surge in BA.5 infection was due to a decreased neutralising antibody response, induced by vaccination and/or natural infection, to neutralize VoC of SARS-CoV-2. It was shown earlier that BA.5 variant infection results in milder pathological manifestations with low lung viral load and attenuated pulmonary pathologies in hamster and hACE2 transgenic mice[18,38]. However, there is a lack of studies that characterize the BA.5-induced

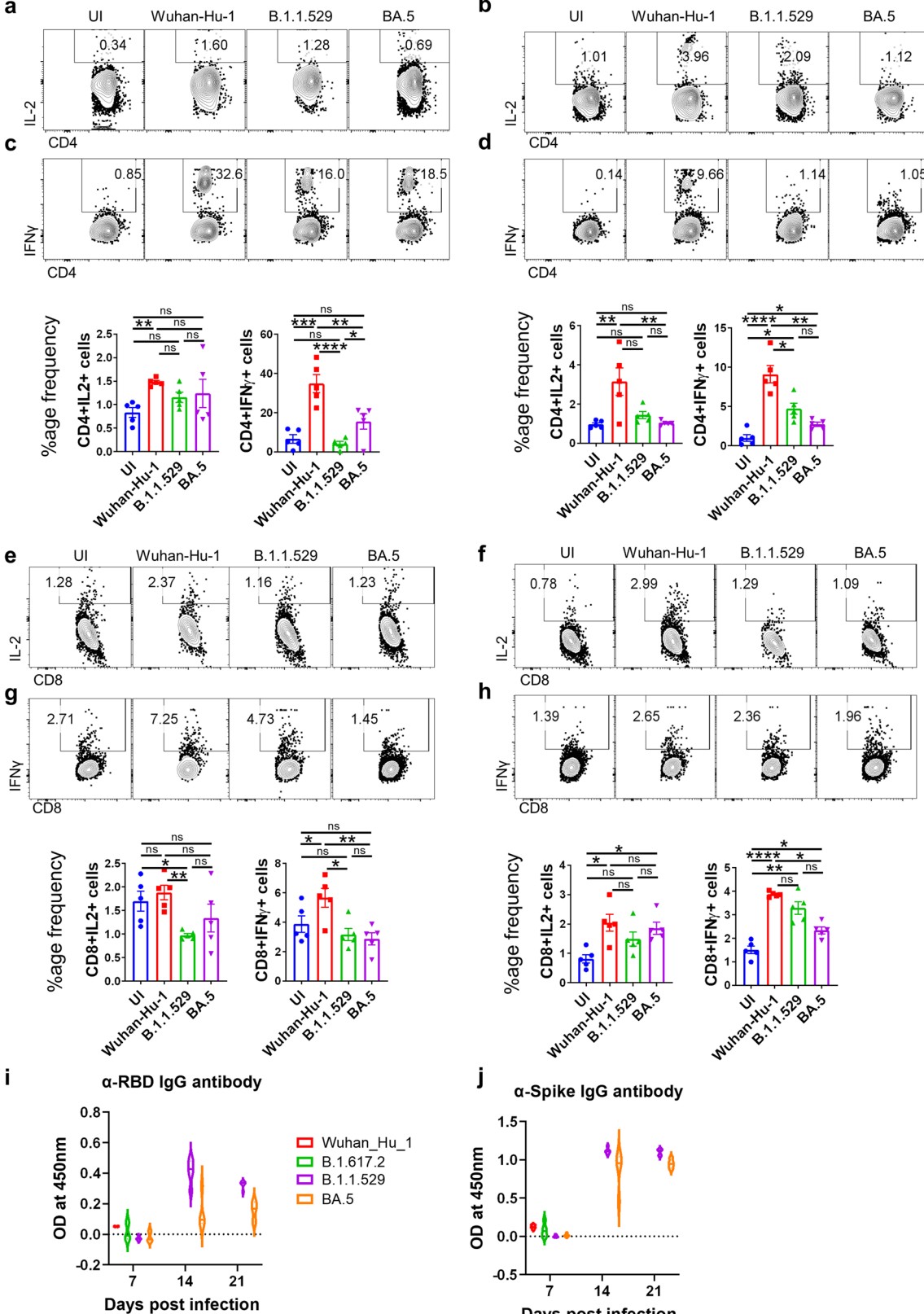

**Fig. 3 Antigen specific humoral and T helper cell response in BA.5 infected mice.** Assessment of immunogencity of BA.5 infection against wild type RBD protein or spike protein was done by evaluating the cellular T cell effector response. Splenocytes from the infected mice at day 6 post infection were harvested and in-vitro stimulated with (**a**, **c**, **e**, **g**, **i**) WT-RBD protein and (**b**, **d**, **f**, **h**, **j**) spike protein then assessed for (**a–d**) CD4$^+$ T cell activation and Th1 response as well as (**e–h**) CD8$^+$ T cell response. Dot plot and bar graph showing % frequency of (**a**, **b**) IL-2 or (**c**, **d**) IFN-γ-producing CD4$^+$ T cells and (**e**, **f**) IL-2 or (**g**, **h**) IFN-γ-producing CD4$^+$ T cells. Humoral response was evaluated through ELISA for (**i**) anti-RBD IgG antibody titer and (**j**) anti-Spike protein IgG antibody titer for 3 time points post infection viz day 7, day 14 and day 21 post infection. For all experiment $n = 5$. One way-Anova using non-parametric Kruskal–Wallis test for multiple comparison. ns non-significant, *$P < 0.05$, **$P < 0.01$, ***$P < 0.001$, ****$P < 0.0001$.

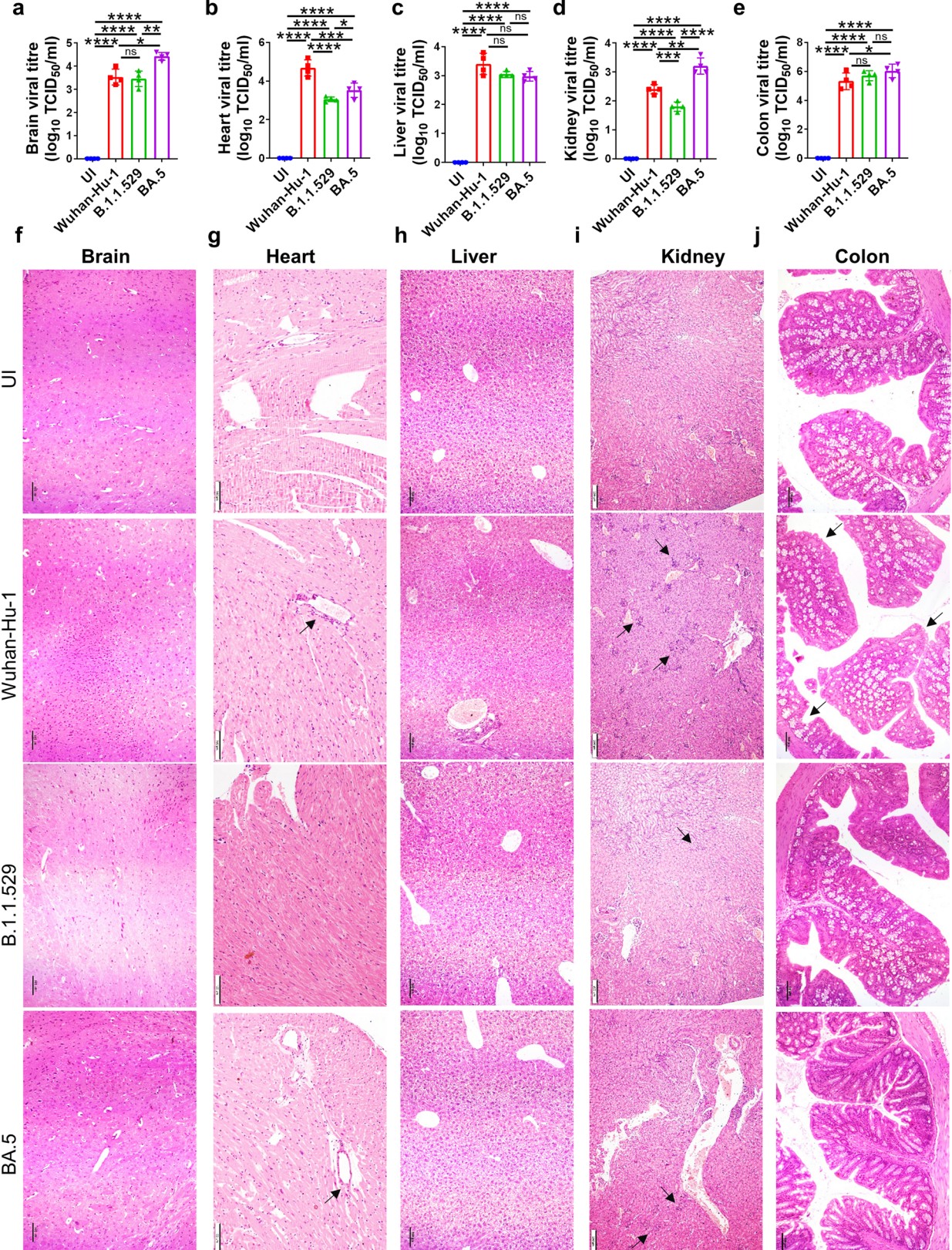

**Fig. 4 BA.5 infection of extra-pulmonary organs.** The virus locatization and histological changes in the major extra-pulmonary organs were studied on 6 days post infection upon necropsy through HE staining. **a–e** Viral load expressed in terms of log10 TCID50/ ml for brain, heart, liver, kidney and colon respectively. **f–j** Representative H & E stained section of brain, heart, liver, kidney and colon respectively showing inflammation at 10X magnification. Black arrow indicates inflammation. For all experiment $n = 5$. One way-Anova using non-parametric Kruskal–Wallis test for multiple comparison. ns non-significant, ****$P < 0.0001$.

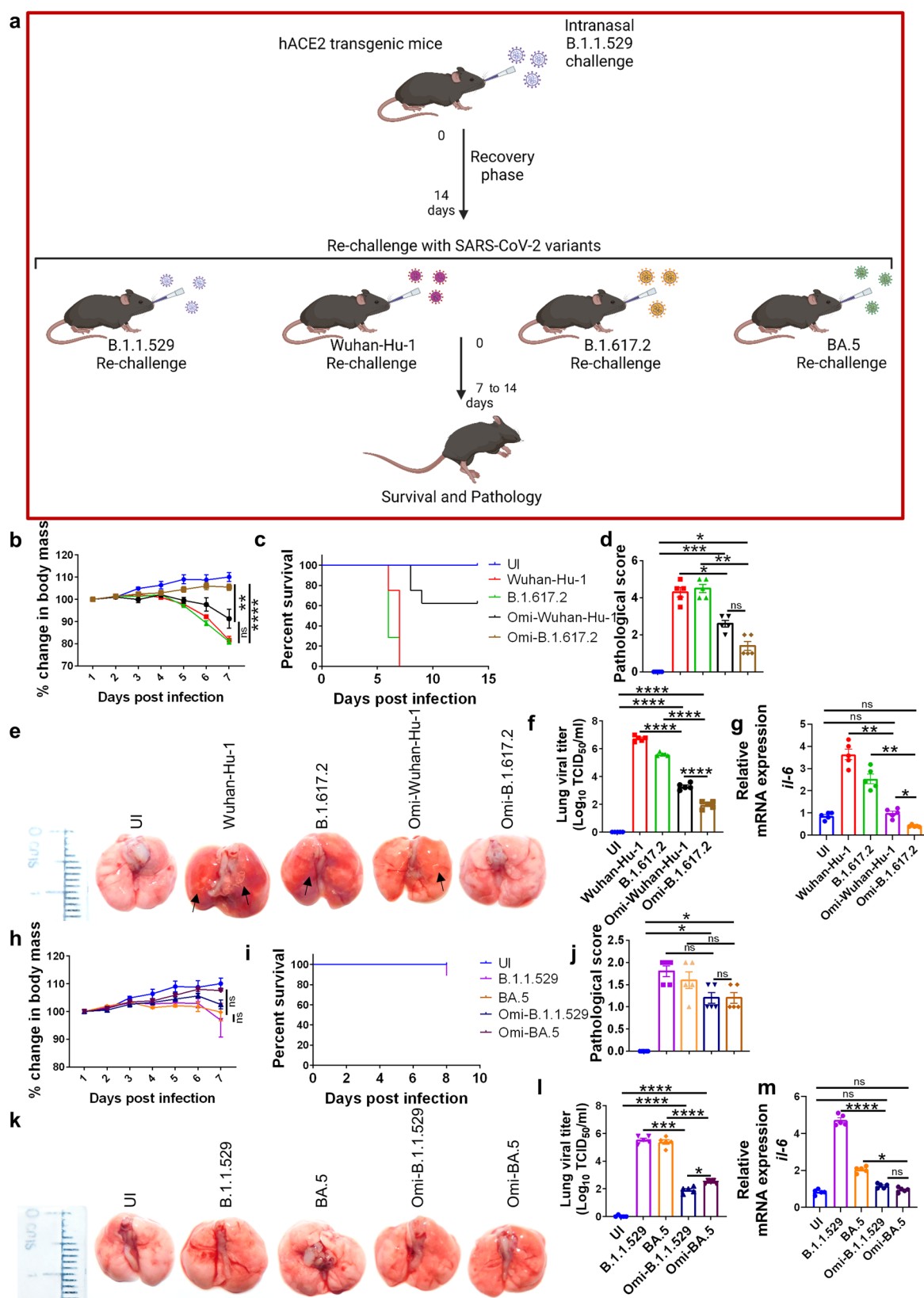

immunopathological manifestations and tissue-specific dissemination to extra-pulmonary organs. In addition, it is unclear whether BA.5 infection induces a protective T cell response, which could be protective against ancestral and other VoC of SARS-CoV2. To understand these points, we used hACE2.Tg mice which were, previously, used to study acute SARS-CoV-2

infection[12,13,15,16]. One of the other advantages of using hACE2.Tg mice are readily available reagents to characterize immunopathological changes and other immunological responses, especially T cell responses.

We first characterized comparative pulmonary manifestations induced by BA.5, ancestral, and B.1.1.529 variant infection.

**Fig. 5 Prior infection with Omicron protects hACE2.Tg mice from BA.5 challenge.** 20 hACE2 transgenic mice were given omicron (B.1.1.529) infection and were allowed to recover. After 14 days mice were divided into 4 groups: Omi-Wuhan-Hu-1 (Omicron recovered-rechallenged with Wuhan-Hu-1 strain), Omi-B.1.617.2 (Omicron recovered-rechallenged with Delta variant), Omi-B.1.1.529 (Omicron recovered-rechallenged with Omicron variant), Omi-BA.5 (Omicron recovered-rechallenged with BA.5 sub-variant) and the severity of reinfection was studied and compared with the control infected or uninfected mice. **a** Scheme showing the experimental design for rechallenge study. **b** %age change in body mass of Omi-Wuhan-Hu-1 & Omi-B.1.617.2 mice and its (**c**) percentage survival as compared to the Wuhan-Hu-1 or B.1.617.2 infection alone. **d**, **e** Representative images of excised lungs showing regions of inflammation & pneumonitis (black arrow) and pathological score (**f**) viral load in the lungs and (**g**) relative levels of mRNA expression of *il-6* gene in the lung samples from the same group. **h–m** Clinical changes observed in omi-B.1.1.529 and omi-BA.5 mice as compared to B.1.1.529 or BA.5 infection alone group. **h** %age changes in the body mass (**i**) percentage survival (**j** and **k**) representative images of the excised lungs and pathological score (**l**) viral laod (**m**) relative mRNA expression of *il-6* from the lung samples. For all experiment *n* = 5. One way-Anova using non-parametric Kruskal–Wallis test for multiple comparison. ns non-significant, *$P < 0.05$, **$P < 0.01$, ***$P < 0.001$, ****$P < 0.0001$.

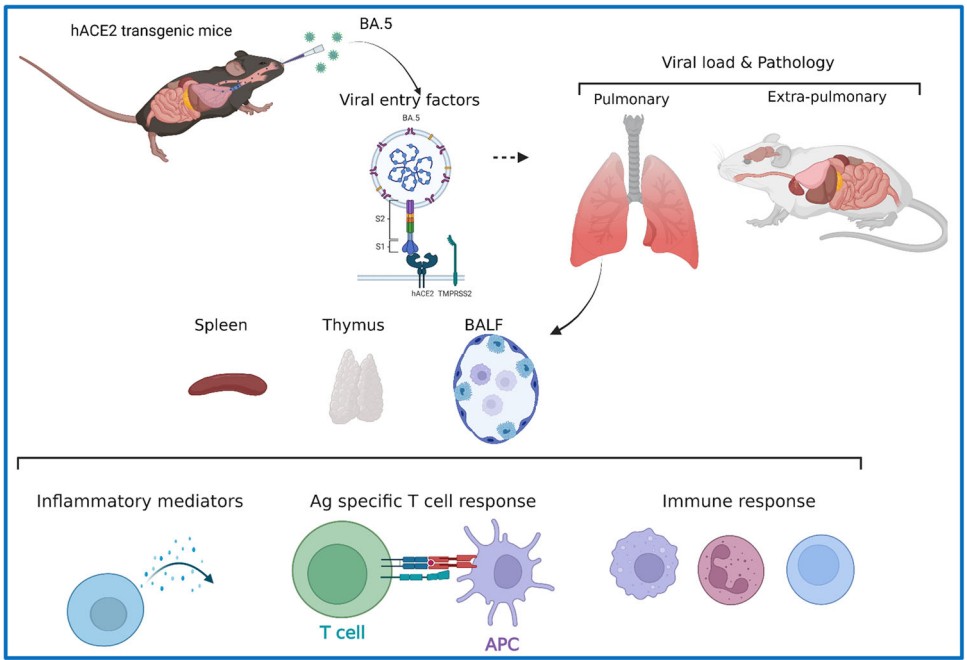

**Fig. 6 Summary figure illustrating the study design and results.** The summary figure of the study indicating the experimental design and important findings. The immunopathology of BA.5 infection (as compared to ancestral Wuhan strain and parental lineage Omicron strain) was studied in hACE2 transgenic mice with the aim to understand the changes in the inflammatory response, humoral and T cell response and the viral load tropism in extra-pulmonary organs.

Consistent with the pathological data from the hamster study, we show a similar milder pulmonary manifestation upon BA.5, as compared to ancestral, infection in hACE2.Tg mice. This is in line with earlier studies with B.1.1.529, BA.1, BA.2 variants, which have shown milder pulmonary manifestations[5,6,9,10]. However, infection-induced inflammatory response post-BA.5 infection is not well characterized. We used a series of approaches to investigate the extent of inflammatory cytokine production and the anti-viral response, which could be useful in understanding the attenuated pulmonary pathology and immune response in BA.5 infection ([10,11,32,38]). Interestingly, we show high levels of mRNA expression of TNF-α and IFN-γ, and frequency of eosinophils and mast cells in BA.5-infected hACE2.Tg mice. These mediators were shown to influence the severity of COVID-19 in a prominent number of hospitalized cases[39–43].

Viral load data of extra-pulmonary organs shows a preferential accumulation of BA.5 in brain, kidney, colon and spleen of hACE2.Tg mice, as TCID50 of BA.5 in these organs were significantly higher than ancestral Wuhan-Hu-1 strain. Yet the ancestral strain is known to cause severe infection pathology as compared to BA.5 infection. This could be due to the acquired adaptation of BA.5 variant to infect extrapulmonary organs dependent on other entry factors than hACE2 receptor leading to

lesser pulmonary pathology. In our opinion, this could be further investigated in a separate study. In addition, whether BA.5 infection in extrapulmonary organs could result in the dysfunction of these organs also warrants further investigation. Though our histopathological analysis does reveal a higher infiltration with an increased pathological score in the brain and kidney of BA.5-infected animals, indicating that BA.5 infection may lead to both neurological and nephrological complications.

Furthermore, we evaluated whether BA.5 infection leads to immunity, especially T cell correlates of protection, that can protect mice from reinfection with ancestral and other variants of SARS-CoV2. It is shown that neutralizing antibody responses were compromised in controlling Omicron and its sublineages including BA.5 ([6,10,32]). Our data support the previous notion that COVID19 vaccination-induced immunity is attenuated against Omicron or its sub-lineage strain infection. In line with this, our data suggest that Omicron and BA.5, as compared to ancestral, infection induced a diminished T cell response in addition to a poorer neutralizing antibody response. These findings, together, suggest that ancestral Wuhan-based vaccines may not be as effective in neutralizing Omicron and its sub-lineage variant infection.

Omicron variant took over delta variant of SARS-CoV2 in terms of circulation due to its high transmissibility (ref).

However, it is not celar whether Omicron-infected individuals will remain immune to other VoC of SARS-CoV2. To understand this, we developed Omicron recovery model severity of the infection in Omiron-recoverd hACE2.Tg mice were tested for other VoC of SARS-CoV2. Our rechallenge study data demonstrated the immunopathological basis of reduced disease severity in Omicron-recovered mice rechallenged with SARS-CoV-2 variants. It was interesting to note that the reinfection with ancestral Wuhan-Hu-1 strain resulted in about 25–30% mortality and subtle manifestations of pulmonary pathologies in Omicron-recovered mice, however, a more lethal Delta variant reinfection showed complete protection. A probable reason for this observation could be the phylogenetic closeness of B.1.617.2 and B.1.1.529 as compared to Wuhan-Hu-1 and B.1.1.529 spike protein makeup[44–46]. Taken together, we provide the first detailed immunopathological characterization of BA.5 infection by using hACE2.Tg mice model. Further, the viral tissue tropism and a possible link to extra-pulmonary pathology was investigated. In addition, we provide T cell response data for BA.5 infection against WT RBD/ Spike protein and a pathological insight into the re-infection model. This may prove important for the hybrid vaccination strategy for countering the emerging variants such as BA.5 sub-lineage variant infection.

## Methods

**Animals**. K18-humanized ACE2 transgenic mice (Strain #:034860, B6.Cg-Tg(K18-ACE2)2Prlmn/J) were originally procured from Jackson laboratory and maintained in a conventional pathogen free environment under 12 h light and dark cycle and fed a standard pellet diet and water ad libitum at small animal facility (SAF) at Translational Health Science and Technology Institute (THSTI)[13]. Institutional Animal Ethics Committee (IAEC) THSTI approved the protocols and procedures (approval number: IAEC/THSTI/217). We have complied with all relevant ethical regulations for animal testing. Protocols related to infection in animals were approved by Institutional Biosafety Committee (IBSC) THSTI and Review Committee on genetic manipulation (RCGM) and Standard Operating Procedure (SOP) for SARS-CoV-2 infection study.

**Antibodies**. Anti-mouse monoclonal antibodies were used for flow cytometry accessement: α-CD3-BV510 (145-2C11, Biolegend, 1:600), α-CD4-PerCp-Cy5.5 (GK1.5, Biolegend, 1:1000), α-CD8-BV410 (53-6.7, Biolegend, 1:1000), α-γδTCR FITC (UC7-13D5, Biolegend, 1:500), α-NK1.1 APC (S17016D, Biolegend, 1:700), α-IFN-γ-PE (XMG1.2, Biolegend, 1:400), IL17A-APC (TC11-18H10, Biolegend, 1:400), α-IL-2-PE-Cy7 (11B11, Biolegend, 1:400), IL-10-FITC (JES5-16E3, Biolegend, 1:400), α-Gr1-BV421 (RB6-8C5, Biolegend, 1:1500), α-CD11b PerCp-Cy5.5 (#101228, Biolegend, 1:1500), α-c-kit BV510 (2B8, Biolegend, 1:800), α-FcЄr1α-APC (MAR-1, Biolegend, 1:500).

**Virus preparation and determination of viral titers**. SARS-Related Coronavirus 2, Isolate USA-WA1/2020, Isolate hCoV-19/ USA/PHC658/2021 (Delta Variant) and SARS-CoV-2 B.1.1.529 variant (Omicron) virus, BA.5 strain was used as challenge strain and was grown and titrated in Vero E6 cell line grown in Dulbecco's Modified Eagle Medium (DMEM) complete media containing 4.5 g/L D-glucose, 100,000 U/L Penicillin-Streptomycin, 100 mg/L sodium pyruvate, 25 mM HEPES and 2% FBS. The virus stocks were plaque purified and amplified at THSTI Infectious Disease Research Facility (Biosafety level 3 facility) as described previously[13].

**SARS-CoV-2 infection**. hACE2-Tg transgenic mice intranasal infection with SARS-CoV-2 was carried out as previously described[12,13]. Briefly, the infected group animals were given injectable anesthesia ketamine (100–150 mg/kg) and xylazine (5–10 mg/kg) and live pre-titrated SARS-CoV-2 or its variants were intranasally administered at $10^5$ PFU (50 μl/ mice) inside the ABSL3 facility. All the animals were monitored till their recovery from anesthesia. The infection protocol was approved by IBSC and RCGM.

**Disease symptoms of SARS-CoV-2 infection**. Mice from infected and uninfected groups were housed for approx. 6-7 days post-challenge for studying immunopathology and for 10–14 days for studying survival at ABSL3. The body mass and activity parameters were recorded daily from the day of the challenge. At the endpoint of the study, animals were euthanized and a necropsy was performed. Serum was collected through a direct blood draw by cardiac puncture. Organs including lungs, spleen and draining lymph nodes (dLN) were collected for imaging and further analysis. Bronchio-alveolar lavage fluid (BALF) was collected by flushing the lungs of the mice, through a tracheal incision, with 0.5 ml phosphate buffer saline (PBS). The left lower lobe of the lung was fixed in 10% formalin and used for histological analysis. The remaining lung was homogenized in Trizol and used for RNA isolation. Spleen and dLN were strained through 40 μm cell strainer with the help of a syringe plunger and single cell suspension (SCS) obtained was used for immunophenotyping or qPCR studies.

**Flow cytometry and intracellular cytokine staining**. SCS obtained from spleen and dLN were either surface stained by using fluorescent anti-mouse antibodies in FACS buffer (PBS with 1% FBS) and analyzed as previously described[47–49]; or were in-vitro in the presence of RBD (2 μg/ml) or its absence (with phorbol 12-myristate13-aceate (PMA; 50 ng/ml; Sigma-Aldrich) and ionomycin (1 μg/ml; Sigma-Aldrich). RBD stimulation was performed for 6 days and was used for intracellular cytokine staining. For PMA + Ionomycin cells were activated for 4 h in the presence of Monensin (#554724 Golgi-Stop, BD Biosciences). All the cells were first washed and blocked with Fc block (anti-mouse CD16/32, Biolegend) at room temperature (RT) for 20 min. Thereafter, cells were used for surface staining with α-CD3, α-CD4, α-CD8, α-CD11b, α-NK1.1, α-Gr1, α-c-kit, α-γδTCR, α-FcЄr1α for 20 min at RT in dark. Thereafter, cells were fixed in Cytofix and permeabilized with Perm/Wash Buffer using a Fixation Permeabilization solution kit (#554714, BD Biosciences). Permeabilized cells stained intracellular cytokine antibodies by using -IFNγ, α-IL17A, α-IL10 antibodies in a permeabilizing buffer in the dark for 20 min at RT. The acquisition of the flow cytometry data was done on (Canto II; BD Bioscience) and analyzed on FlowJo software (Tree-Star).

**Measurement of viral load**. RNA isolated from lung or other organ samples homogenized in Trizol (Invitrogen) was used for viral load estimation. Quantitation of isolated RNA was done and 1 μg of total RNA was then reverse-transcribed to cDNA using the iScript cDNA synthesis kit (Biorad; #1708891) (Roche). 1:5 diluted cDNAs were then used for qPCR by using KAPA SYBR® FAST qPCR Master Mix (5X) Universal Kit (KK4600) along with the on a Fast 7500 Dx real-time PCR system (Applied Biosystems) and the results were analyzed with SDS2.1 software. Briefly, cDNA was used as a template for the CDC-approved commercial reagent for SARS-CoV-2 N gene: 5′-GACCCCAAAATCAGCGAAAT-3′ (Forward), 5′-TCTGGTTACTGCCAGTTGAATCTG-3′ (Reverse). Beta-actin gene was used as an endogenous control and was used for

normalization through quantitative RT-PCR. The region of N gene of SARS-CoV-2 starting from 28287 – 29230 was cloned into pGEM®-T-Easy vector (Promega). This clone was linearized using SacII enzyme and in vitro transcribed using the SP6 RNA polymerase (Promega). The transcript was purified and used as a template for generating a standard curve to estimate the copy number of SARS-CoV-2 N RNA as previously described[50,51].

**qRT-PCR**. For qPCR from splenocytes or lung samples, RNA isolation and cDNA synthesis was done as described above. Diluted cDNAs (1:5) were then used for qPCR by using KAPA SYBR® FAST qPCR Master Mix (5X) Universal Kit (KK4600) on a Fast 7500 Dx real-time PCR system (Applied Biosystems) and the results were analyzed with SDS2.1 software. Beta-actin gene was used as an endogenous control and was used for normalization through quantitative RT-PCR. The relative gene expression was calculated by -ΔΔCt formula as described previously[47,52,53]. The following primer sets were used:

Il-13 – 5'-CTTAAGGAGCTTATTGAGGAG-3' 3'- CATTGCA ATTGGAGATGTTG-5';; Oas2 – 5'-TTATAAAATACCGGCAG CTC-3' 3'-ATTACAGGCCTCTTTTTCTG-5'; Oas3 – 5'-CCAAA CTTAAGAGCCTGATG-3' 3'-GCCTCTCCTCCTTTATATCG-5'; RNasel – 5'-ATACTGTAGGTGATCTGCTG-3' 3'-AAGTAT CTCCTTCATTCCCC-5'; CCCTACTCATAAAAATCACCAG-3' 3'-TTGGAATAGCATTTCCACAG-5'; Ifitm3– 5'-AAGAATCA AGGAAGAATATGAGG-3' 3'-GATCCCTAGACTTCACGG-5'; β-Actin- 5'-TTAATTTCTGAATGGCCCAG-3' 3'-GACCAAAG CCTTCATACATC-5'; Il-17 – 5'-ACGTTTCTCAGCAAACTTA C-3' 3'-CCCCTTTACACCTTCTTTTC-5'; Ifn-γ – 5'-TGAGTAT TGCCAAGTTTGAG-3' 3'-CTTATTGGGACAATCTCTTCC-5'; hACE2–5'-TCCATTGGTCTTCTGTCACCCG-3'; 3'AGACCAT CCACCTCCACTTCTC-5'.

**Anti-RBD ELISA**. Direct ELISA for detecting anti-RBD antibodies was performed from day 7 or day 14 sera samples as previously described[12,47]. Briefly, WT-RBD protein (2 μg/ml) was coated overnight in coating buffer in 96 well high-affinity ELISA plates (Nunc) at 4 °C. Next day, 5 % skimmed milk (blocking buffer) was used for blocking and then sera samples from the infected mice were added to the RBD bound ELISA plates at different dilutions and incubated at room temperature (RT) for 1 h. Next, three washing steps with wash buffer (PBS + 0.05 % tween 20) were performed followed by incubation with anti-mouse IgG antibody detection antibody conjugated with biotinylated (1:1000 dilution) for 1 h. The wells were subsequently washed and incubated further with the enzyme Avidin-HRP (1:10000 dilution) (Sigma) for 30 min at RT. Finally, wells were washed five times and TMB substrate (Thermo Fisher Scientific) 50 μl was incubated in the dark for 15–20 min at RT. The enzyme reaction was stopped by adding 50 μl of 1 N $H_2SO_4$. OD was recorded at 450 nm in a microtiter plate reader.

**Histology**. The left lower lobe of the excised lung from each mouse were used for histopathological analysis by HE staining and Immunohistochemistry (IHC) as previously described[54,55]. Briefly, organ samples were paraffin embedding and cut into 2-μm-thick sections and then mounted on glass slides. With one section HE staining was done, while with the other section IHC for N-antigen using monoclonal antibody against N-protein was done. Stained sections were visualized at 20X magnification and assessment was carried out by random blinded scoring method by a trained pathologist. The scoring was based on the following scale of 0–5 (where 0 meant no observable pathological feature and score of 5 was given to highest pathological feature).

**Re-challenge study**. Re-challenge model for Omicron infection was developed by re-infecting the hACE2.Tg mice after 14 days of the first challenge with the Omicron variant. The first challenge with the omicron variant was given intranasally as described above under injectable anesthesia. Animals showing no disease signs and high RBD antibody titer were then used for re-challenge with different VoC at $10^5$ pfu/ mice. Post-challenge, the animals were observed for clinical signs of infection and changes in body mass. The animals were then sacrificed on day 6-7 post rechallenge and accessed for immunophenotypical changes.

**Statistics and reproducibility**. Graph pad prism 7.0 software was used for the analysis and plotting of all datasets. Body mass, flow cytometry, qPCR, ELISA, and histological score data were compared and analyzed using one-way ANOVA with multiple comparisons as mentioned in the legend. Each experiment was independently repeated 3 times. *P*-value of less than 0.05 and was considered as statistically significant.

**Reporting summary**. Further information on research design is available in the Nature Portfolio Reporting Summary linked to this article.

### Data availability

All the datasets generated in this study are either added as the supplementary figure or have been deposited in online at Dryad. The raw data, figure files and the tables deposited in Dryad: https://doi.org/10.5061/dryad.47d7wm3kt.

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

## Acknowledgements

Financial support to the AA laboratory from THSTI core, Translational Research Program (TRP), BIRAC grants (BT/CS0054/21 and BT/CTH/0004/21) Department of Biotechnology (DBT) and DST-SERB. ZAR is supported by intramural funding (THSTI). This study is partly supported by CEPI through the Project titled "Multi-epitope, Nanoparticle based broadly protective Beta coronavirus candidate vaccine". Immunology Core and FACS facility for providing support in experimentation BA.5 sub-lineage was sequenced by Dr. Bhabatosh Das and isolated in Bioassay Lab. We acknowledge SAF and the infectious disease research facility (IDRF) for their support. ILBS bio-bank for support in histologal analysis and assessment. RCB microscopy facility for microscopic examination of the histology slide. We acknowledge the technical support of Manas and Sandeep. SARS-CoV-2 and its variants were deposited by the Centres for Disease Control and Prevention and obtained through BEI Resources, NIAID.

## Author contributions

Conceived, and supervised the study: A.A., P.K.G.; Designed the study: A.A., Z.A.R.; Animal experiment at ABSL3: Z.A.R., K.S., N.A.; FACS: Z.A.R., S.Sadhu, N.A., V.D.; qPCR: N.A., K.S., Z.A.R.; ELISA: J.D.; Microscopy: V.S., V.D.; Source of virus: Anil.K.P., S.B.; Virus propagation, titration and sequencing: R.K., J.S., G.M., S.M., S.Samal, B.D., Anil.K.P., Amit.K.P.; Data analysis & compilation: Z.A.R.; Contributed reagents/materials/analysis tools: A.A.; Wrote & revised the manuscript: Z.A.R., A.A.

## Competing interests

The authors declare no competing interest.
