## [Peer Review File · Communications Biology]

Reviewers' comments:

Reviewer #1 (Remarks to the Author):

In this manuscript from Rizvi et al., the authors have examined the consequences of SARS-CoV-2 BA.5 infection in hACE2 transgenic mice. Infection with the Wuhan-Hu-1 strain and the parental Omicron B1.1.529 strains were used as comparators. BA.5 like the parental Omicron strain causes less pathology in this model, similar to studies on Omicron in the hamster model. Prior infection with Omicron was protective against BA.5 (and against delta).

It is not clear why only figures are in the body of the text, but the legends are not. This is not a very helpful way of presenting data.

While these are solid studies and I am generally supportive of this work I have two concerns, of which the first is easier to resolve.

1. Figure 5 with re-challenge data during Omicron convalescence is the most impressive part of this study. However, the title of the paper is totally confusing and disagrees with the abstract. The abstract clearly states that prior infection with B1.1.529 protects against BA.5 That is not what the title says. This is in keeping with the writing overall of this otherwise excellent manuscript- grammar and syntax is frequently thrown to the winds.
2. In Figure 3 the logic of using only the RBD domain protein for T cell responses was lost on me. Vaccines use the entire Spike protein. What is the relevance of only looking at T cell responses to the RBD domain? Conserved sequences in the entire Spike protein may generate T cell responses.

Reviewer #2 (Remarks to the Author):

Comments:

Rizvi et. al. investigated the immunopathology as well as the cellular T cell response in the lung and spleen upon Omicron BA.5 sublineage infection using the K18-hACE2 transgenic mouse model. In addition, evidences hinting the extended tissue tropism caused by BA.5 infection were provided. The author also tempted to investigate the cross-protection contributed by primary infection with Omicron against other variants. The data on inflammation/antiviral response in the lung and the draining lymph node immune cell landscape is interesting. However, there is a considerable amount of discrepancies remain between the observations and conclusions. I also have concerns on the presented data in a number of the panels.

Major:

1. Figure 1e and Suppl Figure 6b. Image from the uninfected group is not an IHC image.
2. Figure 1g, 1h, 2f, 4a-e. The variations are too small to be real, particularly considering they should be retrieved from different animals.
3. TNF α and IFN γ were previously shown to increase tissue damage and animal mortality upon SARS-CoV-2 infection (Karki, R., et al., 2021). In the current manuscript, BA.5 infection led to aberrantly high induction of these cytokines yet the infection outcomes (tissue damage and animal survival) were improved.
4. Infection of the central nervous system was demonstrated to be associated with fatal infection outcome of the K18-hACE2 transgenic mice infected by SARS-CoV-1/SARS-CoV-2 (McCray, P.B., Jr., et al., 2007; Fumagalli, V., et al., 2021). The author showed viral N gene copies were increased in multiple extrapulmonary organs including the brain of the BA.5-infected hACE2 Tg mice yet survival of the BA.5 infected animals were significantly higher than those infected by wildtype SARS-CoV-2, which had lower viral gene copies in the brain. These data seem to be contradictory. Besides, further evidence (i.e. detection of viral protein, ds vRNA, infectious titres) should be provided to validate the replicative virus infection in those pulmonary organs.

5. The author claims prior infection with Omicron provided "broader" protection against BA.5 rechallenge. However, data shown in figure 5 can only show that primary infection with BA.1 led to similar protection against both BA.1 and BA.5. Therefore the data provided here was not sufficient to support the claim.

Minor:

1. Line 83: L452R instead of L542R.
2. Line 82-84: BA.4 and BA.5 shared the identical amino acids in the spike.
3. Line 86: Wildtype SARS-CoV-2 also harbors the D61 amino acid in ORF6. Therefore, it does not help to explain the BA.5 wave of infection.
4. Line 330: The author claims molecular basis for attenuated pulmonary pathology caused by BA.5 was provided in the current study. It's an overstatement?
5. Figure 5a. For the wildtype- and Delta-infected mice, survival was monitored until day 14. Therefore the schematic illustration should be revised to match the data.
6. Line 304. "Severity" was misspelled.

Reviewer #3 (Remarks to the Author):

The authors describe pathogenicity of the Omicron subvariant BA.5 in transgenic hACE2 mice. They found that BA.5 had similar replication compared to B.1.1.529 along with lung pathology. The authors also examined the induction of cytokines and interferon-stimulated genes by RT-PCR. In some cases, BA.5 infection significantly increased or decreased certain genes compared to mock infection or Wuhan infection. The authors also looked at re-infection.

There are certain issues in the manuscript and this reviewer feels conclusions are drawn without the proper data.

- infectious virus needs to be evaluated by plaque assay; qPCR is not sufficient
- the hACE2 mouse and hamster study of BA.5 in Nature Uraki R et al should be cited
- changes in gene expression is not sufficient; protein levels of cytokines/chemokines in the lung homogenates by bioplex analysis should be performed
- the re-infection study was performed ONLY 14 days after the primary infection; this is not long enough to generate any conclusions
- line 249, again the authors claim to "show an overall waning T cell immunity for BA.5 infected" the authors never give a time frame that they restimulated splenocytes after the primary infection
- parameters such as viral loads and induction of cytokines should be examined at more than one time point

Reviewers' comments:

Reviewer #1 (Remarks to the Author):

In this manuscript from Rizvi et al., the authors have examined the consequences of SARS-CoV-2 BA.5 infection in hACE2 transgenic mice. Infection with the Wuhan-Hu-1 strain and the parental Omicron B.1.1.529 strains were used as comparators. BA.5 like the parental Omicron strain causes less pathology in this model, similar to studies on Omicron in the hamster model. Prior infection with Omicron was protective against BA.5 (and against delta).

It is not clear why only figures are in the body of the text, but the legends are not. This is not a very helpful way of presenting data.

While these are solid studies and I am generally supportive of this work I have two concerns, of which the first is easier to resolve.

Re: We thank the reviewer for the in-depth review, critical comments and valuable suggestions in improving the manuscript. We have now revised the manuscript substantially with new datasets in line with the suggestions made by the reviewer.

1. Figure 5 with re-challenge data during Omicron convalescence is the most impressive part of this study. However, the title of the paper is totally confusing and disagrees with the abstract. The abstract clearly states that prior infection with B.1.1.529 protects against BA.5 That is not what the title says. This is in keeping with the writing overall of this otherwise excellent manuscript- grammar and syntax is frequently thrown to the winds.

Re: We appreciate the reviewers for their valuable suggestions in improving the quality of the manuscript. In line with the suggestions made by the reviewer, we have revised the title of the manuscript to "Omicron sub-lineage BA.5 infection causes attenuated pathology with compromised ancestral SARS-CoV-2 specific T and B cell response in hACE2 transgenic mice". In addition, we have also worked on the grammar and syntax of the manuscript and have tried to improve it wherever required.

2. In Figure 3 the logic of using only the RBD domain protein for T cell responses was lost on me. Vaccines use the entire Spike protein. What is the relevance of only looking at T cell responses to the RBD domain? Conserved sequences in the entire Spike protein may generate T cell responses.

Re: Few subunit vaccines against COVID-19 have relied on the formulation involving RBD domain and that was the reason we wanted to see the T cell response against BA.5 against ancestral RBD protein. However, reviewers made a very valid point of using the entire Spike protein as spike protein are crucial for host-pathogen interaction and have been reported to accumulate major mutations leading to diminished immunological response. In line with the suggestions made by the reviewer, we have now included results on T cell response against both RBD protein and spike protein (as Figure 3). In addition, we have also included antibody response against these two proteins.

Reviewer #2 (Remarks to the Author):

Comments:

Rizvi et. al. investigated the immunopathology as well as the cellular T cell response in the lung and spleen upon Omicron BA.5 sublineage infection using the K18-hACE2 transgenic mouse model. In addition, evidences hinting the extended tissue tropism caused by BA.5 infection were provided. The author also tempted to investigate the cross-protection contributed by primary infection with Omicron against other variants. The data on inflammation/ antiviral response in the lung and the draining lymph node immune cell landscape is interesting. However, there is a considerable amount of discrepancies remain between the observations and conclusions. I also have concerns on the presented data in a number of the panels.

Major:

1. Figure 1e and Suppl Figure 6b. Image from the uninfected group is not an IHC image.

Re: We thank the reviewer for the valuable comment and for helping us to improve the manuscript. The images from the uninfected group are IHC images counterstained with HE. Since uninfected group will not have any N-protein antigen, these slides do not take any stain of IHC. Hence, the uninfected control group appears as HE stained image. These are negative controls for IHC and do not change the overall IHC assessment of the infection control groups.

2. Figure 1g, 1h, 2f, 4a-e. The variations are too small to be real, particularly considering they should be retrieved from different animals.

Re: Thank you for asking this question. Data shown in 1g, 1h, 2f, 4a-e (previous numbering) of Log10 viral load in organs of infected animals corroborates with the viral load data trend from previously published literature. These hACE2 transgenic mice show consistent virus entry and viral load owing to hACE2 receptor expression. Moreover, it should also be noted that the y-axis of the viral load data set is expressed in log10 and not linear scale.

Winkler, E.S., Bailey, A.L., Kafai, N.M. *et al.* SARS-CoV-2 infection of human ACE2-transgenic mice causes severe lung inflammation and impaired function. *Nat Immunol* **21**, 1327–1335 (2020). <https://doi.org/10.1038/s41590-020-0778-2>

Uraki, R., Halfmann, P.J., Iida, S. *et al.* Characterization of SARS-CoV-2 Omicron BA.4 and BA.5 isolates in rodents. *Nature* **612**, 540–545 (2022). <https://doi.org/10.1038/s41586-022-05482-7>

Halfmann, P.J., Iida, S., Iwatsuki-Horimoto, K. *et al.* SARS-CoV-2 Omicron virus causes attenuated disease in mice and hamsters. *Nature* **603**, 687–692 (2022). <https://doi.org/10.1038/s41586-022-04441-6>

3. TNF α and IFN γ were previously shown to increase tissue damage and animal mortality upon SARS-CoV-2 infection (Karki, R., et al., 2021). In the current manuscript, BA.5 infection led to aberrantly high induction of these cytokines yet the infection outcomes (tissue damage and animal survival) were improved.

Re: The reviewers correctly pointed the fact that both TNF α and IFN γ are strongly correlated with pulmonary injury during COVID-19 and high levels of both of these cytokines are implicated in worsening of the diseases. In our BA.5 challenged lung tissues we observed an increase in both TNF α and IFN γ as compared to the uninfected lung samples. It is worthy to

note that though IFN γ levels in infected lungs was high as compared to the uninfected lungs, it was significantly lower than the IFN γ mRNA expression in Wuhan infected lung samples suggesting that the overall IFN γ mediated pulmonary injury would be lower in BA.5 as compared to the Wuhan infection. Important to note that both TNF α and IFN γ cytokines are crucial for anti-viral response and their regulated increase maybe important for viral clearance as seen in BA.5 infected lungs showing attenuated viral load and less pulmonary pathology as compared to Wuhan infected lungs.

van der Ploeg K, Kiro Singh AS, Mori DAM, Chakraborty S, Hu Z, Sievers BL, Jacobson KB, Bonilla H, Parsonnet J, Andrews JR, Press KD, Ty MC, Ruiz-Betancourt DR, de la Parte L, Tan GS, Blish CA, Takahashi S, Rodriguez-Barrquer I, Greenhouse B, Singh U, Wang TT, Jagannathan P. TNF- α ⁺ CD4⁺ T cells dominate the SARS-CoV-2 specific T cell response in COVID-19 outpatients and are associated with durable antibodies. *Cell Rep Med*. 2022 Jun 21;3(6):100640. doi: 10.1016/j.xcrm.2022.100640. Epub 2022 May 3. PMID: 35588734; PMCID: PMC9061140.

Todorović-Raković N, Whitfield JR. Between immunomodulation and immunotolerance: The role of IFN γ in SARS-CoV-2 disease. *Cytokine*. 2021 Oct;146:155637. doi: 10.1016/j.cyto.2021.155637. Epub 2021 Jul 3. PMID: 34242899; PMCID: PMC8253693.

4. Infection of the central nervous system was demonstrated to be associated with fatal infection outcome of the K18-hACE2 transgenic mice infected by SARS-CoV-1/SARS-CoV-2 (McCray, P.B., Jr., et al., 2007; Fumagalli, V., et al., 2021). The author showed viral N gene copies were increased in multiple extrapulmonary organs including the brain of the BA.5-infected hACE2 Tg mice yet survival of the BA.5 infected animals were significantly higher than those infected by wildtype SARS-CoV-2, which had lower viral gene copies in the brain. These data seem to be contradictory. Besides, further evidence (i.e. detection of viral protein, ds vRNA, infectious titres) should be provided to validate the replicative virus infection in those pulmonary organs.

Re: Thank you for raising this very relevant question. This is an interesting paradigm that we observed during BA.5 challenge study. The overall pathophysiology of BA.5 infection mice was less severe as compared to ancestral Wuhan strain infection as has been shown previously by other groups. However, when we evaluated viral load in the brain we found higher viral load as compared to the brain viral load in the Wuhan infected. This unusual high viral load in the brain, now also validated with TCID50 virus titre data (incorporated as new figure 4a), was also recently reported by Stewart et al. in which they showed that BA.5 infection is highly neuroinvasive. Remarkably, they showed that BA.5 infection is also lethal in K18-hACE2 transgenic mice, however they showed this by using a different dose and route as compared to the one reported in our manuscript.

Uraki, R., Halfmann, P.J., Iida, S. et al. Characterization of SARS-CoV-2 Omicron BA.4 and BA.5 isolates in rodents. *Nature* 612, 540–545 (2022). <https://doi.org/10.1038/s41586-022-05482-7>

Stewart R, Ellis SA, Yan K, Dumenil T, Tang B, Nguyen W, Bishop C, Larcher T, Parry R, Sullivan RKP, Lor M, Khromykh AA, Meunier FA, Rawle DJ, Suhrbier A. Omicron BA.5 infects human brain organoids and is neuroinvasive and lethal in K18-hACE2 mice. *bioRxiv* 2022.12.22.521696; doi: <https://doi.org/10.1101/2022.12.22.521696>

5. The author claims prior infection with Omicron provided “broader” protection against BA.5 re-challenge. However, data shown in figure 5 can only show that primary infection with BA.1

led to similar protection against both BA.1 and BA.5. Therefore, the data provided here was not sufficient to support the claim.

Re: We thank the reviewer for critically evaluating our manuscript and providing insightful comments to improve our manuscript. We have toned down the interpretation in the result section and have moderated it as suggested by the reviewers.

Minor:

1. Line 83: L452R instead of L542R.

Re: Thank you. It is corrected now.

2. Line 82-84: BA.4 and BA.5 shared the identical amino acids in the spike.

Re: We have corrected the statement and have made it more coherent.

3. Line 86: Wildtype SARS-CoV-2 also harbours the D61 amino acid in ORF6. Therefore, it does not help to explain the BA.5 wave of infection.

Re: We thank the reviewer for the comment and improving the quality of the manuscript. We have now revised the statement in order to make it more coherent for the readers.

4. Line 330: The author claims molecular basis for attenuated pulmonary pathology caused by BA.5 was provided in the current study. It's an overstatement?

Re: We agree with the suggestions made by the reviewer. We have now moderated the statement in line with the suggestions.

5. Figure 5a. For the wildtype- and Delta-infected mice, survival was monitored until day 14. Therefore, the schematic illustration show be revised to match the data.

Re: Thank you for bringing this to our notice. We have now corrected the schematic illustration.

6. Line 304. "Severeity" was misspelled.

Re: Thank you. We have corrected it.

Reviewer #3 (Remarks to the Author):

The authors describe pathogenicity of the Omicron subvariant BA.5 in transgenic hACE2 mice. They found that BA.5 had similar replication compared to B.1.1.529 along with lung pathology. The authors also examined the induction of cytokines and interferon-stimulated genes by RT-PCR. In some cases, BA.5 infection significantly increased or decreased certain genes compared to mock infection or Wuhan infection. The authors also looked at re-infection.

There are certain issues in the manuscript and this reviewer feels conclusions are drawn without the proper data.

-infectious virus needs to be evaluated by plaque assay; qPCR is not sufficient

Re: Thank you, we have now added the TCID50 data for viral load evaluation. The ambiguous qPCR data has been completely removed from the manuscript.

-the hACE2 mouse and hamster study of BA.5 in Nature Uraki R et al should be cited

Re: Thank you. We have now cited Uraki R. et al hamster and hACE2 transgenic mice study of BA.5 in our manuscript.

-changes in gene expression is not sufficient; protein levels of cytokines/chemokines in the lung homogenates by bioplex analysis should be performed

Re: We thank the reviewer for this suggestion. We have now included antigen specific (both Spike protein and RBD protein) cytokine response for Wuhan, Omicron and BA.5 variants through intracellular flow cytometry (New Fig. 3). However, we could not perform the bioplex analysis due to unavailability of resources.

-the re-infection study was performed ONLY 14 days after the primary infection; this is not long enough to generate any conclusions

Re: We thank the reviewer for the insightful comment and helping us improve the quality of the manuscript. For the re-challenge study, we chose day 7 and 14 post re-challenge as the end point for the study to evaluate lung viral load and survival respectively. The mice re-challenged with Wuhan/ Delta strain died within 14 days and therefore could not be examined further. For the Omicron/ BA.5 re-challenge study we observed very low virus titre (1-2 on log10 scale) at day 7 post infection and highly diminished pulmonary pathology. Since the primary aim of this experiment was to evaluate the pulmonary viral load and pathology and since we observed attenuated pathology in re-infection model of Omicron and BA.5, we reason that increasing the time kinetics will not have any overall significant changes in the results related to pulmonary viral load and pathology.

-line 249, again the authors claim to “show an overall waning T cell immunity for BA.5 infected” the authors never give a time frame that they re-stimulated splenocytes after the primary infection

Re: Thank you for pointing this missing information from the manuscript. The splenocytes were from the infected mice were harvested at 7 days post infection and was used for restimulation with spike or RBD protein in-vitro to study T cell response. This information has also been incorporated in the manuscript.

-parameters such as viral loads and induction of cytokines should be examined at more than one time point

Re: We thank the reviewer for suggesting us to study the time dependent progression of BA.5 infections and the subsequent viral load and cytokine response. We have now included

lung viral load and the humoral immune response data for multiple time points post infection as suggested (Fig 1I and Fig 3i & 3j).

Reviewers' comments:

Reviewer #1 (Remarks to the Author):

The experimental work done by Rizvi et al., is solid and while the data should be reported, the writing appears to get sloppier with each revision, and some of the newly highlighted portions are quite meaningless and very hard to understand. The corresponding author must take care to read what is being put out. A strong experimental effort is compromised by garbled writing here and there.

1. Just a few examples here. This work needs to be carefully edited so that the reader can understand what the authors are trying to say.

For example, the new highlighted line below in the revised Discussion makes no logical or scientific sense. Pango lineage is a hierarchical tree-like classification system. What are the authors referring to here?

"Line 360: Our data supports the previous notion that Pango variant, because of its accumulated spike mutation, results in attenuated cellular T cell response against wild type RBD and spike protein restimulation. This result may find further implication in booster dose regime for hybrid immunity against Pango lineage"

Many minor issues similar to those below:

Line 242: "is even more aggravated" What is meant?

Line 341: "surprisingly we did unexpected found high levels of"

2. Given the many sequence changes in RBD and NTD sequences between the Omicron strain as compared to the Wuhan strain the results in Figure 3 are not particularly surprising. As a result, the current "new" title is a bit misleading. There really is not an established "compromised ancestral SARS-CoV-2 specific T and B cell response" but possibly also an absence of some degree of cross-reactivity.

A more accurate and simple title would be:

"Omicron sub-lineage BA.5 infection results in attenuated pathology in hACE2 transgenic mice"

Many Minor issues (too many to list):

For example: Figure 3 legend:

Please change "accessment" to "assessment".

Please change "was harvested" to "were harvested"

Reviewer #2 (Remarks to the Author):

1. IHC for the uninfected control group should be performed. The purpose of a control is to see if there is unspecific staining in the uninfected samples using the lab's staining protocol. Using a H&E in place of IHC is NOT acceptable.

2. Please show the specific data points in all panels. Does one data point represent one individual mouse? Should make it very clear in the figure legends.

3. The authors are seeing more virus in the brain but less pathologies. This is different from the observation of Stewart R et al. Please explain why.

Reviewer #3 (Remarks to the Author):

The authors have sufficiently addressed my concerns.

Reviewers' comments:

Reviewer #1 (Remarks to the Author):

The experimental work done by Rizvi et al., is solid and while the data should be reported, the writing appears to get sloppier with each revision, and some of the newly highlighted portions are quite meaningless and very hard to understand. The corresponding author must take care to read what is being put out. A strong experimental effort is compromised by garbled writing here and there.

1. Just a few examples here. This work needs to be carefully edited so that the reader can understand what the authors are trying to say.

For example, the new highlighted line below in the revised Discussion makes no logical or scientific sense. Pango lineage is a hierarchical tree-like classification system. What are the authors referring to here?

“Line 360: Our data supports the previous notion that Pango variant, because of its accumulated spike mutation, results in attenuated cellular T cell response against wild type RBD and spike protein restimulation. This result may find further implication in booster dose regime for hybrid immunity against Pango lineage”

Re: We greatly appreciate the comments and suggestions made by the reviewers which have helped us in improving the overall quality of the manuscript and its readability.

Many minor issues similar to those below:

Line 242: “is even more aggravated” What is meant?

Line 341: “surprisingly we did unexpected found high levels of”

Re: We thank the reviewer for their valuable input. We have now thoroughly and carefully gone through the entire manuscript and have corrected the grammar and syntax errors in the manuscript. The above two ambiguities in line 242 and line 341 have been revised and corrected.

2. Given the many sequence changes in RBD and NTD sequences between the Omicron strain as compared to the Wuhan strain the results in Figure 3 are not particularly surprising. As a result, the current “new” title is a bit misleading. There really is not an established “compromised ancestral SARS-CoV-2 specific T and B cell response” but possibly also an absence of some degree of cross-reactivity.

A more accurate and simple title would be:

“Omicron sub-lineage BA.5 infection results in attenuated pathology in hACE2 transgenic mice”

Re: We are grateful to the reviewer for their valuable input in helping us to improve the manuscript quality. We thank the reviewer for the manuscript title suggestion, we accept the suggested title as the new title of the manuscript.

Many Minor issues (too many to list):

For example: Figure 3 legend:

Please change “accessment” to “assessment”.

Please change “was harvested” to “were harvested”

Re: We thank the reviewers for their valuable efforts in helping us to improve the manuscript's quality and readability. We have corrected the Figure 3 legend as per the suggestion. We have also now thoroughly gone through the entire manuscript and have corrected the grammatical and syntax error in the manuscript.

Reviewer #2 (Remarks to the Author):

1. IHC for the uninfected control group should be performed. The purpose of a control is to see if there is unspecific staining in the uninfected samples using the lab's staining protocol. Using a H&E in place of IHC is NOT acceptable.

Re: We thank the reviewer for their valuable input. We have now put new IHC control images for Fig 1e and Supplementary fig 6b & 6d.

2. Please show the specific data points in all panels. Does one data point represent one individual mouse? Should make it very clear in the figure legends.

Re: We are grateful to the reviewers for their comments which have helped us in improving the manuscript quality. We have now replaced all the figure files showing summary data with the individual data set figures throughout the figure file. Each data set represented on the graph is representative of the individual mouse sample from the experiment. The number of mice used in each experiment has been incorporated in the figure legend.

3. The authors are seeing more virus in the brain but less pathologies. This is different from the observation of Stewart R et al. Please explain why.

Re: We are thankful to the reviewer for their query on the brain pathology in BA.5 infected hACE2.Tg mice. The BA.5 infected hACE2.Tg mice showed the highest brain viral load (among all the SARS-CoV-2 strains) as detected by TCID50 data. The HE-stained section of the brain from BA.5 also showed a more pathological score as compared to the parental B.1.1.529 strain (omicron). However, we found the brain pathological score of the ancestral Wuhan-Hu-1 strain more as compared to BA.5 infected brain. This finding corroborates well with the Stewart R (biorxiv, 2022) data showing increased neuroinvasion and pathology of BA.5 infections in hACE2.Tg mice. In line with our findings, they showed high neuro-infection by BA.5 as compared to Wuhan-Hu-1 (ancestral) in hACE2.Tg mice. They further carried out the detailed neuropathological characterization of the changes in the brain of BA.5 infected hACE2.Tg mice. Though we did not carry out detailed neuropathology of brain sections in BA.5 infected mice, as it was not the focus area of the current study, our pathological score data did point out increased pathology in BA.5 infected mice as compared to Omicron infected mice which are in line with the Stewart R data.

Moreover, to improve the readability of the results, we have revised the result describing the pathological score of brain in BA.5 infected mice (lines 282-285).

Reviewer #3 (Remarks to the Author):

The authors have sufficiently addressed my concerns.

Re: We are grateful for the reviewer's valuable comment and suggestions which has contributed to the improvement of overall manuscript quality.

Reviewers' comments:

Reviewer #1 (Remarks to the Author):

The experimental work done by Rizvi et al., is solid and while the data should be reported, the writing appears to get sloppier with each revision, and some of the newly highlighted portions are quite meaningless and very hard to understand. The corresponding author must take care to read what is being put out. A strong experimental effort is compromised by garbled writing here and there.

1. Just a few examples here. This work needs to be carefully edited so that the reader can understand what the authors are trying to say.

For example, the new highlighted line below in the revised Discussion makes no logical or scientific sense. Pango lineage is a hierarchical tree-like classification system. What are the authors referring to here?

“Line 360: Our data supports the previous notion that Pango variant, because of its accumulated spike mutation, results in attenuated cellular T cell response against wild type RBD and spike protein restimulation. This result may find further implication in booster dose regime for hybrid immunity against Pango lineage”

Re: We greatly appreciate the comments and suggestions made by the reviewers which have helped us in improving the overall quality of the manuscript and its readability.

Many minor issues similar to those below:

Line 242: “is even more aggravated” What is meant?

Line 341: “surprisingly we did unexpected found high levels of”

Re: We thank the reviewer for their valuable input. We have now thoroughly and carefully gone through the entire manuscript and have corrected the grammar and syntax errors in the manuscript. The above two ambiguities in line 242 and line 341 have been revised and corrected.

2. Given the many sequence changes in RBD and NTD sequences between the Omicron strain as compared to the Wuhan strain the results in Figure 3 are not particularly surprising. As a result, the current “new” title is a bit misleading. There really is not an established “compromised ancestral SARS-CoV-2 specific T and B cell response” but possibly also an absence of some degree of cross-reactivity.

A more accurate and simple title would be:

“Omicron sub-lineage BA.5 infection results in attenuated pathology in hACE2 transgenic mice”

Re: We are grateful to the reviewer for their valuable input in helping us to improve the manuscript quality. We thank the reviewer for the manuscript title suggestion, we accept the suggested title as the new title of the manuscript.

Many Minor issues (too many to list):

For example: Figure 3 legend:

Please change “accessment” to “assessment”.

Please change “was harvested” to “were harvested”

Re: We thank the reviewers for their valuable efforts in helping us to improve the manuscript's quality and readability. We have corrected the Figure 3 legend as per the suggestion. We have also now thoroughly gone through the entire manuscript and have corrected the grammatical and syntax error in the manuscript.

Reviewer #2 (Remarks to the Author):

1. IHC for the uninfected control group should be performed. The purpose of a control is to see if there is unspecific staining in the uninfected samples using the lab's staining protocol. Using a H&E in place of IHC is NOT acceptable.

Re: We thank the reviewer for their valuable input. We have now put new IHC control images for Fig 1e and Supplementary fig 6b & 6d.

2. Please show the specific data points in all panels. Does one data point represent one individual mouse? Should make it very clear in the figure legends.

Re: We are grateful to the reviewers for their comments which have helped us in improving the manuscript quality. We have now replaced all the figure files showing summary data with the individual data set figures throughout the figure file. Each data set represented on the graph is representative of the individual mouse sample from the experiment. The number of mice used in each experiment has been incorporated in the figure legend.

3. The authors are seeing more virus in the brain but less pathologies. This is different from the observation of Stewart R et al. Please explain why.

Re: We are thankful to the reviewer for their query on the brain pathology in BA.5 infected hACE2.Tg mice. The BA.5 infected hACE2.Tg mice showed the highest brain viral load (among all the SARS-CoV-2 strains) as detected by TCID50 data. The HE-stained section of the brain from BA.5 also showed a more pathological score as compared to the parental B.1.1.529 strain (omicron). However, we found the brain pathological score of the ancestral Wuhan-Hu-1 strain more as compared to BA.5 infected brain. This finding corroborates well with the Stewart R (biorxiv, 2022) data showing increased neuroinvasion and pathology of BA.5 infections in hACE2.Tg mice. In line with our findings, they showed high neuro-infection by BA.5 as compared to Wuhan-Hu-1 (ancestral) in hACE2.Tg mice. They further carried out the detailed neuropathological characterization of the changes in the brain of BA.5 infected hACE2.Tg mice. Though we did not carry out detailed neuropathology of brain sections in BA.5 infected mice, as it was not the focus area of the current study, our pathological score data did point out increased pathology in BA.5 infected mice as compared to Omicron infected mice which are in line with the Stewart R data.

Moreover, to improve the readability of the results, we have revised the result describing the pathological score of brain in BA.5 infected mice (lines 282-285).

Reviewer #3 (Remarks to the Author):

The authors have sufficiently addressed my concerns.

Re: We are grateful for the reviewer's valuable comment and suggestions which has contributed to the improvement of overall manuscript quality.